# Evolutionary Divergence and Radula Diversification in Two Ecomorphs from an Adaptive Radiation of Freshwater Snails

**DOI:** 10.3390/genes13061029

**Published:** 2022-06-08

**Authors:** Leon Hilgers, Stefanie Hartmann, Jobst Pfaender, Nora Lentge-Maaß, Ristiyanti M. Marwoto, Thomas von Rintelen, Michael Hofreiter

**Affiliations:** 1LOEWE Centre for Translational Biodiversity Genomics (TBG), 60325 Frankfurt, Germany; 2Institute of Biochemistry and Biology, Universität Potsdam, 14476 Potsdam, Germany; stefanie.hartmann@uni-potsdam.de (S.H.); michael.hofreiter@uni-potsdam.de (M.H.); 3Museum für Naturkunde, 10115 Berlin, Germany; nora.lentge-maass@mfn.berlin (N.L.-M.); thomas.vonrintelen@mfn.berlin (T.v.R.); 4Naturkundemuseum Potsdam, 14467 Potsdam, Germany; jobst.pfaender@rathaus.potsdam.de; 5Museum Zoologicum Bogoriense, Research Centre for Biosystematics and Evolution, National Research and Innovation Agency, Cibinong 16912, Indonesia; ristiyanti456@gmail.com

**Keywords:** speciation, adaptive radiation, molluscs, RNAseq, regulatory evolution, trophic specialization

## Abstract

(1) Background: Adaptive diversification of complex traits plays a pivotal role in the evolution of organismal diversity. In the freshwater snail genus *Tylomelania*, adaptive radiations were likely promoted by trophic specialization via diversification of their key foraging organ, the radula. (2) Methods: To investigate the molecular basis of radula diversification and its contribution to lineage divergence, we used tissue-specific transcriptomes of two sympatric *Tylomelania sarasinorum* ecomorphs. (3) Results: We show that ecomorphs are genetically divergent lineages with habitat-correlated abundances. Sequence divergence and the proportion of highly differentially expressed genes are significantly higher between radula transcriptomes compared to the mantle and foot. However, the same is not true when all differentially expressed genes or only non-synonymous SNPs are considered. Finally, putative homologs of some candidate genes for radula diversification (hh, arx, gbb) were also found to contribute to trophic specialization in cichlids and Darwin’s finches. (4) Conclusions: Our results are in line with diversifying selection on the radula driving *Tylomelania* ecomorph divergence and indicate that some molecular pathways may be especially prone to adaptive diversification, even across phylogenetically distant animal groups.

## 1. Introduction

Adaptive radiations provide extreme examples of rapid phenotypic and ecological diversification and, therefore, feature prominently among model systems for adaptation and speciation [1,2,3,4,5]. In many adaptive radiations, lineage divergence was promoted by the diversification of a few key adaptive traits, such as foraging organs [3,6,7,8,9,10]. Understanding the genetic basis of such key adaptive traits is essential because they shape the evolutionary trajectories of diversifying lineages [10,11,12]. Previous work indicates that polygenic selection [13,14,15], adaptive introgression [16,17,18,19], and regulatory evolution [14,17,20,21] promote diversification in adaptive radiations [13,14,15]. However, much remains to be discovered about the genetic basis of adaptive traits, the molecular evolution underlying their diversification, and their contribution to speciation [2,22]. Here we use sympatric ecomorphs of *T. sarasinorum* to investigate the genetic basis of the diversification of the molluscan key-foraging organ (the radula) and its role in lineage divergence in a radiation of freshwater snails [23].

The genus *Tylomelania* is endemic to the central Indonesian island Sulawesi and underwent several radiations following the colonization of different lake systems [24,25]. Lacustrine species flocks occur across heterogeneous substrates and exhibit a remarkable radula diversity (Figure 1) [26,27]. In contrast, riverine clades occupy relatively homogenous substrates, have uniformly shaped radular teeth, and include comparatively few species [26,27]. Additionally, some radula morphologies likely evolved independently on similar substrates in different lakes [24,25,27]. Hence, it was hypothesized that divergent adaptation of the radula allowed efficient foraging on alternative substrates and promoted speciation in radiations of *Tylomelania* [24,25]. In addition to interspecific variation, some species exhibit radula polymorphisms [27]. One such species is *T. sarasinorum*, which reportedly has a substrate-correlated radula polymorphism. Ecomorphs occur on rocks and logs in the shallow waters of Lake Towuti (Figure 1 and Figure 2a) [27] but cannot be distinguished based on mitochondrial markers [24,28]. Given the radula’s hypothesized role as a key adaptive trait in this radiation, ecomorphs may represent diverging lineages adapted to alternative foraging substrates (Figure 2a). Novel rows of radula teeth are built continuously throughout life [29] and tooth shapes can be altered rapidly in some snails [30]. However, phenotypic plasticity of *T. sarasinorum* radulae appears unlikely as both ecomorphs occur on both substrates yet changes in radula morphology across teeth rows have never been observed in ~500 specimens (von Rintelen, unpublished data).

To measure morphological and genetic divergence of sympatric *T. sarasinorum* ecomorphs, we combine morphological analyses of the radula and the shell with tissue-specific transcriptomes. Our results indicate evolutionary divergence of ecomorphs, support the hypothesis that the radula acts as the key adaptive trait, and suggest that morphological diversification in adaptive radiations is achieved via a limited set of conserved signaling pathways.

## 2. Materials and Methods

**Specimen and tissue collection:** Adult *T. sarasinorum* were collected from submerged wood and rock substrates at Loeha Island (Lake Towuti, Sulawesi, Indonesia; 2.76075° S 121.5586° E). Snails were collected in close proximity to each other and kept in buckets with lake water for several hours before dissection. Tissue samples were stored in RNAlater, radula-forming tissue was separated from the remaining radula (Appendix A), and radula morphs were inspected with scanning electron microscopy (SEM) using a ZEISS EVO LS 10 instrument. Voucher specimens are stored at Museum Zoologicum Bogoriense, Cibinong, Indonesia (MZB Gst. 12.366).

**Morphological analyses of the shell:** Shell shape and radula meristic were assessed for 37 specimens from the collection of the Natural History Museum Berlin that had been sampled randomly from wooden (*n* = 19) and rocky substrates (*n* = 18), (Appendix A). Radula morphs were determined based on previously published descriptions. If the central denticle of the rachis was enlarged with a blunt edge compared to the other denticles, specimens were classified as rock morph [27,31]. Differences in abundances of radula morphs between substrates were determined with a χ^2^-test. Variation in shell shape was quantified using geometric morphometrics. Specimens were photographed with the aperture facing upwards using a SatScan (SmartDrive Limited). Landmarks and semi-landmarks were placed on the whorls and aperture (Appendix A) using tpsDIG2 [32]. Differences in size and rotation were removed from the data with a Procrustes superimposition and a principal component analysis (PCA) was calculated on the Procrustes residuals using geomorph [33].

**Morphological analyses of the radula:** Radulae were dissected and tissue was digested in 500 µL lysis buffer with 10 µL proteinase K at 55 °C [34]. Radulae were cleaned with an ultrasound bath and sputter-coated with a Quorum Q150RS Sputter Coater. Teeth were counted and maximum width and length of the central denticle and width of the rachis base were measured using ImageJ [35] (Appendix A). Ratios of denticle width/total height and rachis width were calculated. A PCA was carried out with these ratios and the number of denticles on the rachis. Two tailed t-tests were used to evaluate morphological differences between ecomorphs, based on principal components (PCs) capturing at least 5% of the total variance.

**Sample preparation and sequencing**: Nineteen wood morph specimens were grouped into three pools of five and one pool of four individuals (used in Hilgers et al. (2018) [36]), and 20 individuals of the *T. sarasinorum* rock morph into four pools of five individuals (Appendix A for an overview). Tissue samples were weighed (Mettler AT 261) and similar amounts of each individual were pooled resulting in four biological replicates of each tissue. Tissue was homogenized with a Precellys Minilys. RNA was extracted from foot muscle with a TRIzol^®^ extraction according to the manufacturer’s protocol. To extract RNA from minute amounts of radula and mantle, a customized protocol of the RNeasy Plus Micro Kit (Qiagen) was employed [36]. Briefly, remaining tissue fragments were digested with proteinase K following mechanical homogenization. Subsequently, lysis buffer was added for efficient DNA removal with gDNA spin-columns. Amount and quality of total RNA was inspected using Agilent’s 2100 Bioanalyzer. Sequenced samples showed no signs of degradation or DNA contamination. Messenger RNA was enriched using NEXTflex™ Poly-A Beads and strand-specific libraries were built using the NEXTflex™ Rapid Illumina Directional RNA-Seq Library Prep Kit (Bioo Scientific) with modifications as suggested by Sultan et al. (2012) [37]. Quality and DNA concentration of libraries were evaluated using Agilent’s 2100 Bioanalyzer and qPCR (Kapa qPCR High Sensitivity Kit). Libraries were sequenced (150 bp, paired end) on an Illumina NextSeq at the Berlin Center for Genomics in Biodiversity Research. Samples of both ecomorphs were always prepared and sequenced together.

**Transcriptome assembly**: Raw sequences were trimmed (phred-score ≥30, ≥25 bases) and terminal Ns removed using sickle [38] (Appendix A). Adapter sequences were removed with cutadapt [39], generating a final dataset of 941 million paired-end reads (Appendix A). Trinity_v2.1.1 [40,41] was run in strand-specific mode with a minimal transcript length of 250 bp, in silico read normalization (max. read coverage = 50), and twofold minimal kmer coverage to generate a single assembly from all tissues of both ecomorphs. Quality-filtered, adapter-trimmed reads of each sample were mapped to the transcriptome using bowtie2 [42], followed by abundance estimation with RSEM [43]. Since abundance of rRNA mostly reflects polyA capture success, rRNA was removed following identification with a BLAST search using 28S rRNA (*Brotia pagodula*; HM229688.1) and 18S rRNA (*Stenomelania crenulata*; AB920318.1) as query sequences. Pool1 mantle of both ecomorphs and pool1 of the rock morph were identified as outliers in PCA of log_2_ transformed counts per million mapped reads (cpm) (Appendix A). The cause for this is a combination of the lower yield of total RNA in the first extractions resulting in lower library complexity and deeper sequencing of pool1 (Appendix A). A batch effect might also have contributed to this observation because pool1 mantle and pool1 radula were sequenced separately (Appendix A) and the corresponding foot pools did not show similar patterns. All samples of pool1 were excluded from further analyses to retain a balanced dataset. The assembly was filtered by expression (FPKM ≥ 1 on gene level: one mapped fragment per kilobase of gene per million mapped reads; isoforms ≥ 5% of gene expression), using the *filter_low_expr_transcripts.pl* script from Trinity. CD-HIT_v4.6 [44] was used to cluster the longest isoforms of all “trinity genes” based on sequence similarity (≥97% sequence identity; ≥90% alignment coverage of shorter sequence). The longest transcript of each cluster was retained. Reads of both ecomorphs were re-mapped to the remaining transcripts, and low expression genes (FPKM < 1) were removed to create a final assembly. BUSCO_v1.1b1 [45] was employed to generate estimates of transcriptome completeness, redundancy, and fragmentation by searching for 843 metazoan single-copy orthologs.

**Gene expression analysis**: Gene expression was analyzed using the pipeline in Trinity_v2.1.1 [40,41]. Quality-filtered, adapter-trimmed reads were mapped to the final assembly using bowtie2 [42], followed by abundance estimation with RSEM [43]. Differentially expressed genes (false discovery rate (FDR) ≤ 10^−2^; fold change (FC) ≥ 4); and highly differentially expressed genes (FDR ≤ 10^−10^; FC ≥ 4) were determined with edgeR [46] (Appendix A). Although divergence of the two ecomorphs is low (<0.002% based on fixed SNPs, median F_st_ = 0.14; mean F_st_ = 0.23) and mapping rates are similar across tissues and the ecomorphs (see Appendix A), we tested for the effects of mapping bias by estimating whether genes with alternatively fixed SNPs (af-SNPs) were more likely to be differentially expressed than expected using Fisher’s exact test. Additionally, we tested whether alleles are erroneously assembled as genes resulting in highly differentially expressed genes. We thus used BLASTN to identify highly DE genes with >95% sequence similarity over 50% of their length to other highly DE genes with opposite regulation.

**Annotation**: Transcripts were annotated using Trinotate_v3.0.1. TransDecoder_v3.0.0 (http://transdecoder.github.io, accessed 25 September 2017) was used to predict open reading frames. Transcripts and predicted proteins were blasted against the UniProtKB/Swiss-Prot database (May 2016) using blast+_v2.3.0 [47]. Transmembrane domains were predicted with TmHmm v2.0c [48], signal peptides were predicted with SignalP_v4.1 [49], RNAmmer_v1.2 [50] was used to identify RNAs, and HMMER3_v3.1b2 [51] was used to search for protein family domains in pfam (May 2016). *T. sarasinorum* genes that are mentioned by name in this manuscript were further verified by searching proteins matching *T. sarasinorum* open reading frames in the UniProt database using BLASTX and manually inspecting alignments of the best hits.

**Ecomorph divergence:** PoPoolation2 [52] was used to study population divergence of *T. sarasinorum* ecomorphs (Appendix A). Duplicate reads, reads that did not map as proper pairs, and low-quality alignments (mapping quality < 20) were removed using SAMtools_v1.3 [53] and Picard-Tools_2.12.1 (http://broadinstitute.github.io/picard/, accessed on 3 June 2022). Mappings of different tissues from one pool were merged. To reduce biases in SNP detection caused by variance in gene expression, a uniform coverage of 20x for each pool was generated by subsampling mapped reads (without replacement) and removing sites with a coverage <20x. SNPs were called at a minor allele frequency (MAF) of 10%. SNPs with lower MAF were discarded to remove sequencing errors and uninformative SNPs [54,55]. SNP-wise F_st_ was calculated using PoPoolation2 [52]. Median pairwise F_st_ values were estimated from all SNPs for each pairwise comparison of pools. Median F_st_ and SNP-wise F_st_ distributions between ecomorphs were calculated based on combined pool-wise allele counts resulting in a coverage of 60× (3 pools, 20× coverage/ecomorph). MAF was retained at 10%. Synonymous and non-synonymous mutations were determined for the longest ORF per gene using the *syn-nonsyn-at-position.pl* script in PoPoolation_v1.2.2. Although PoPoolation is not recommended for transcriptome data [52,55], it was successfully employed in numerous studies [56,57,58,59] and consistent estimates for replicates in this study support the validity of our approach.

**Tissue-specific transcriptomic divergence**: To evaluate whether transcriptomic divergence differed between tissues, tissue-wise divergences in gene expression and coding sequences were determined. To this end, the proportions of DE genes (FDR ≤ 10^−2^; FC ≥ 4) or highly DE genes between identical tissues of both ecomorphs (FDR ≤ 10^−10^; FC ≥ 4) were calculated. Genes that were (highly) DE across all tissues were excluded from the analyses (Appendix A). Likewise, the frequency of af-SNPs was determined for genes expressed (FPKM ≥ 1) in each tissue, excluding universally expressed genes. Differences in proportions of differentially expressed genes and frequency of af-SNPs between tissues were evaluated using Fisher’s exact test.

**Candidate gene****identification:** Non-synonymous af-SNPs were used to identify candidate genes for adaptive divergence. Since core assumptions of models for pooled genomic data may be violated by a larger margin of error in allele frequency estimation from pooled RNA [55], studies using pooled transcriptomic data mostly used quantile-based approaches for outlier detection [57,59]. We used the most conservative approach available and only chose alternatively fixed SNPs (F_st_ = 0 in all within-morph comparisons, F_st_ = 1 in all across-morph comparisons; 98.8% percentile). Genes with non-synonymous af-SNPs expressed in radulae of both ecomorphs and genes that were highly DE between radulae of both ecomorphs (FDR ≤ 10^−10^; FC ≥ 4) but not between mantles or foot tissues (FDR ≥ 10^−5^; FC ≤ 4), were chosen as candidates for radula shape divergence (Appendix A).

## 3. Results

### 3.1. Geometric Morphometrics Corroborates a Habitat-Correlated Radula Polymorphism

Although a habitat-correlated radula polymorphism of *T. sarasinorum* has been reported [31] (Figure 2a), it has not been systematically analyzed. To investigate whether ecomorphs are morphologically distinct and exhibit habitat-correlated abundances, we quantified variation in radula and shell morphology of specimens from wood and rock substrates (Appendix A). Abundances of ecomorphs differed significantly between substrates (rock: 79% rock ecomorph; wood: 100% wood ecomorph; *p* = 1.01 × 10^−6^; χ^2^ test). The ecomorphs differed significantly in both the radula (Principal component (PC) 1: *p* < 0.001) and shell shape (PC2: *p* < 0.001). PC1 of radula shape contained 86.3% of the variation in the dataset and clearly separated the *T. sarasinorum* ecomorphs. In comparison, the differences in shell shape were less pronounced as PC2 only explained 11.8% of shell shape variation (Appendix A) and shell morphospaces of both ecomorphs were overlapping along this axis (Figure 2b).

### 3.2. Transcriptome Sequencing and Assembly

To gain insight into the transcriptomic divergence of sympatric *T. sarasinorum* ecomorphs, we sequenced pooled tissue-specific transcriptomes of each mantle, radula formative tissue (Appendix A), and foot tissue from both ecomorphs (Appendix A). From a de novo assembly of combined data of both ecomorphs, we retained 156,685 genes (status assigned by Trinity) with an N50 of 1229 bp, high completeness (BUSCO [45] completeness: 89%), and a low duplication rate (7.5%) [60,61] (Appendix A).

### 3.3. Transcriptome-Wide SNP Data Indicates Evolutionary Divergence of Ecomorphs

Adaptation of the radula to alternative substrates was hypothesized to promote lineage diversification in adaptive radiations of *Tylomelania* [24,25]. Hence, we investigated whether sympatric radula morphs of *T. sarasinorum* with different habitat preferences represent diverging evolutionary lineages. In a total of 39,631,840 bases with coverage (≥20×), we identified 517,825 putative SNPs with a minor allele frequency (MAF) of ≥10%. A total of 6366 SNPs (1.2%) in 2572 transcripts (7.8% of transcripts with SNPs) were alternatively fixed between the ecomorphs (F_st_ = 0 in all within-morph comparisons and F_st_ = 1 in all across-morph comparisons). Although the majority of genetic variation is shared between both ecomorphs at Loeha Island (median F_st_ = 0.14; mean F_st_ = 0.23), we observed an excess of highly differentiated loci and consistently higher F_st_ between pools of different ecomorphs (Figure 3). Although the median F_st_ in the pairwise comparisons among pools of identical ecomorphs ranged from 0.016 to 0.048, it ranged from 0.143 to 0.188 among pools of different ecomorphs, indicating evolutionary divergence of ecomorphs.

### 3.4. Ecomorphs Differ in Gene Expression across all Investigated Tissues

Regulatory evolution resulting in divergent gene expression plays a key role in adaptation and speciation [62,63]. Although gene expression is known to be highly tissue-dependent, much remains to be discovered about tissue-specific transcriptomic divergence and its contribution to speciation [64,65,66]. To investigate gene expression divergence between *T. sarasinorum* ecomorphs, we analyzed the gene expression of the foot, shell-forming mantle, and radula-forming tissue of both ecomorphs. In accordance with previous work, the foot and mantle form sister clusters to the exclusion of the radula (Figure 4b) [36]. Within tissues, samples of different ecomorphs form separate clusters, indicating divergence in gene expression across all investigated tissues (Figure 4a,b). Overall divergence in gene expression was very similar across tissues when we accounted for the number of expressed genes (Figure 4c, Appendix A). Differential gene expression between ecomorphs is not explained by a mapping bias to diverged genes as genes with alternatively fixed SNPs (af-SNPs) were not overrepresented among differentially expressed genes (*p* = 0.92).

### 3.5. Radula Transcriptomes Exhibit Increased Sequence Divergence and Elevated Proportion of Highly DE Genes

Selection experiments revealed rapid tissue-specific transcriptomic divergence in response to selection [56]. We thus expected increased divergence of radula transcriptomes compared to other organs if diversifying selection on the radula drives divergence of *T. sarasinorum* ecomorphs.

Divergence in gene expression was measured as the proportion of expressed genes that were differentially expressed (false discovery rate (FDR) ≤ 10^−2^) or highly differentially expressed (FDR ≤ 10^−10^) between the same tissue types of both ecomorphs. The total number of DE genes and highly DE genes varied considerably among tissues. We found 3025 DE genes (3.5%) in the radula, 6127 DE genes (4.6%) in the mantle, and 6248 DE genes (5.2%) in the foot. The corresponding numbers for highly DE genes were 536 (0.81%) for the radula, 436 (0.34 %) for the mantle, and 424 (0.42 %) for the foot. If alleles were erroneously assembled as genes, they did not dominate among highly DE genes because less than 10% of all highly DE genes had high sequence similarity to highly DE genes with the opposite regulation (radula: 8.2%, mantle: 7.2%, foot: 3%). Genes universally DE (*n* = 1044) or highly DE (*n* = 93) across all tissues are uninformative for estimating tissue-specific divergence and were excluded from the analysis. Divergence in gene expression based on all DE genes is most pronounced in the foot and least pronounced in the radula (Figure 5a,b). However, highly differentially expressed genes are significantly more abundant in the radula than in the mantle (FDR < 10^−10^: 97% higher; *p* < 10^−5^; Fisher’s exact test) and foot (85% higher; *p* < 10^−5^; Figure 5a,b). Of the top 1% and top 5% most significantly differentially expressed genes, 58% and 45%, respectively, were DE between radulae (Top 1% foot: 15%, mantle: 27%; Top 5% foot: 26%; mantle: 29%). Similarly, the proportion of af-SNPs in genes with non-universal expression (FPKM < 1 in at least one tissue) is significantly higher in the radula than in the mantle (~ 34.4% higher in radula, *p* < 10^−5^) or foot (36.6% higher in radula, *p* < 10^−5^; Figure 5b). This pattern remained unchanged when only af-SNPs within open reading frames (ORFs) were considered. No significant differences were found when the analysis was restricted to non-synonymous SNPs (Figure 5b). Finally, significantly more genes were both highly differentially expressed and also carried af-SNPs in the radula compared to the mantle (*p* = 0.0002) and foot (*p* = 0.0095; Figure 5c) and the majority of these were only highly DE between the radulae (25 out of 32; Figure 5c but see Appendix A for lower DE threshold).

### 3.6. Homologs of Candidate Genes for Radula Disparity Contributed to Craniofacial Diversification in Vertebrate Radiations

To investigate genes contributing to radula diversification, two non-overlapping sets of candidate genes were generated. Genes that were highly DE between the radulae of the two ecomorphs (FDR ≤ 10^−10^; FC ≥ 4) but not between mantle or foot tissues (FDR ≥ 10^−5^; FC ≤ 4), were chosen as expression-based candidate genes (*n* = 230). The second set of candidate genes (*n* = 538) was composed of genes that were expressed in the radula of both ecomorphs and carried alternatively fixed non-synonymous SNPs. To further narrow down the list of candidates, we focused on genes involved in gene regulation and cell–cell signaling, because these may determine when and where the radula tooth matrix is secreted.

Although most genes with alternatively fixed non-synonymous SNPs only had one such SNP (66%), a maximum of 12 (plus 10 synonymous ones) was found in Rho GTPase activating protein 21 (*rhg21*) (Appendix A). This corresponds to a 32.2-fold and 2.37-fold increase compared to all analyzed transcripts and all transcripts with af-SNPs, respectively. Rho family GTPase signaling interacts with notch signaling and regulates various cellular functions [67,68,69].

SNP-based candidates further included a transcript annotated as *notch1* and *strawberry notch* (1 non-syn; 5 syn). A putative homolog of *notch1* was also found among the expression-based candidate genes together with the morphogen *hedgehog* (*hh*). Both notch and the hedgehog signaling pathway are conserved across bilaterians and interact during developmental tissue patterning [70,71]. *hh* also played an important role in the evolution of jaw morphology in East African cichlids [72,73,74] and regulates expression of bone morphogenetic proteins (BMPs) in metazoans [71,75,76]. Evolution of craniofacial diversity in both Darwin’s finches and East African cichlids likely involved regulatory evolution of BMPs [21,22,77,78]. Interestingly, *hh* is overexpressed in the radula of the *T. sarasinorum* rock morph, and a bone morphogenetic protein that is most similar to *gbb*/*BMP5-8* is only expressed in the radula of the rock morph.

The *aristaless-like homeobox 1 transcription factor* (*ALX1*) promoted beak diversification in Darwin’s finches [17]. *Aristaless-related homeobox protein* (*arx) is* the only homeobox gene among our candidate genes. In our dataset, *arx* is only expressed in the radula and carries four non-synonymous af-SNPs.

## 4. Discussion

Adaptation of the radula, the foraging organ of molluscs, was hypothesized to promote speciation in adaptive radiations of the snail genus *Tylomelania*. Here, we used two radula morphs of *T. sarasinorum* to investigate the molecular basis of radula disparity and its contribution to speciation in *Tylomelania*.

Our data support the hypothesis that the radula polymorphism of *T. sarasinorum* evolved in adaptation to different foraging habitats. Differences in radula shape were more pronounced than differences in shell shape and abundances of both radula morphs differed significantly between foraging substrates. Our population genomic analyses of transcriptome-wide SNP data indicated evolutionary divergence of sympatric radula morphs of *T. sarasinorum*, which is in line with the hypothesis that radula adaptation to alternative substrates promotes lineage divergence in *Tylomelania*. F_st_ distribution indicated high differentiation of a few genomic regions in a background of shared genetic variation. A scenario that could have given rise to this pattern is divergence with gene flow. During divergence with gene flow, loci under selection become fixed, whereas genomic variation at sites of the genome that are not in strong linkage with selected loci are homogenized by gene flow [79,80]. Individuals with intermediate phenotypes and non-resolving phylogenies from mitochondrial markers indicate gene flow between ecomorphs and between *T. sarasinorum* and other species [27,81]. However, other scenarios, such as divergence without gene flow combined with selective sweeps, may result in similar patterns, albeit with increased absolute divergence in regions that are not linked to outlier loci. Genomic data comprising individuals from other sites and ideally other species would be required to investigate population history and gene flow among divergent lineages to decide between alternative explanations.

Our analyses reveal divergence in gene expression across all investigated tissues, which cannot be explained by mapping bias or alleles that were erroneously assembled as genes. Thus, regulatory evolution appears to contribute to the divergence of *Tylomelania* ecomorphs as indicated by divergent gene expression across all tissues and higher divergence in untranslated than translated regions, both in general and in transcripts of highly DE genes (Figure 5b,c). These findings are in accordance with the expectation that selection favors regulatory change that can avoid deleterious pleiotropic effects [82,83]. Additionally, our results suggest that the divergence of ecomorphs is polygenic, which is in line with results from other study systems. For example, regulatory evolution contributed to ecological divergence in East African cichlids, Darwin’s finches and sticklebacks [2,14,17,84], and polygenic selection gave rise to convergent gene expression in lake whitefish radiations in Europe and North America [63].

Since selection can promote tissue-specific transcriptomic divergence [56], we hypothesized that transcriptomic divergence would be elevated in the radula compared to other tissues, if diversifying selection on the radula drove ecomorph divergence in *T. sarasinorum*. Accordingly, both the proportion of highly differentially expressed genes and the frequency of af-SNPs are significantly higher in genes expressed in the radula, compared to genes expressed in the other investigated tissues. However, this pattern was not detected for all DE genes or non-synonymous SNPs. Since the vast majority of expression differences between species are likely generated by drift and selectively nearly neutral [85,86], a considerable number of DE genes at higher FDR may thus result from drift rather than diversifying selection. Alternatively, the increased divergence of gene expression in the foot based on all DE genes might hint at so far undetected factors that contribute to the divergence of *T. sarasinorum* ecomorphs. A caveat of our study, which is that RNA from foot samples was extracted using a different protocol, may have affected estimates of foot-specific divergence in gene expression. However, this cannot explain the increased sequence divergence in the radula or the increased proportions of highly DE genes in the radula compared to the mantle. Thus, overall, our observations of increased divergence of the radula transcriptome based on highly differentially expressed genes and af-SNPs add support to the hypothesis that diversifying selection on the radula promoted the evolutionary divergence of *T. sarasinorum* ecomorphs.

Finally, we identified candidate genes for radula diversification. With a maximum of 12 non-synonymous SNPs, Rho GTPase-activating protein 21 (*rhg21*) stood out among the SNP-based candidates (Appendix A). Rho GTPase-activating proteins activate Rho family GTPase signaling, which among other functions, regulates cytoskeletal reorganization [67,68,69]. Coordinated reorganization of the cytoskeleton is particularly interesting with respect to the radula polymorphism of *T. sarasinorum* because odontoblasts undergo pronounced shape changes during radula tooth secretion, and the modification of their cell shape likely influences tooth morphology [29]. In addition to changing odontoblast cell shapes, modified cytoskeletons may change the localization of chitin synthesis via altered actin filament guidance of a lophotrochozoan-specific chitin synthase with a myosin head [87] that is expressed in radula-forming tissue [36]. The dramatically increased frequency of non-synonymous af-SNPs per kb of ORF in genes such as *rhg21* (32.2 fold) could be caused by different mechanisms including the accumulation of mutations due to increased mutation rates or relaxed purifying selection, positive selection driving the fixation of non-synonymous mutations, or differences in the time of divergence between genes, where alleles of some genes diverged before *T. sarasinorum* ecomorphs and either persisted as standing genetic variation or introgressed from different lineages. Gene flow among diverging lineages and introgression from more distantly related species is common in adaptive radiations and generates and maintains genetic variation at the loci underlying adaptive traits [2,14,16,17,19,88,89,90]. Since previous studies indicate abundant hybridization among *Tylomelania* [24,27], the extraordinary divergence of a few genes (Appendix A) suggests that selection on introgressed alleles may also contribute to divergence in *Tylomelania* radiations. Genomic data from across the radiation could be used to test this hypothesis, which, if confirmed, would add further support to a combinatorial view of speciation) [91].

Furthermore, putative homologs of genes that contributed to the diversification of beaks in Darwin’s finches and/or the jaws of East African cichlids might also be involved in the adaptive diversification of the radula. These include a bone morphogenetic protein (BMP, specifically *gbb*/*BMP5-8*) and hedgehog (*hh*). Hedgehog regulates BMP expression in several metazoan lineages [71,75,76] and mediates both fixed and phenotypically plastic effects on jaw morphology in East African cichlids [72,73,74]. Regulatory evolution resulting in the divergent expression of BMPs played a pivotal role in craniofacial diversity in both Darwin’s finches and East African cichlids [21,22,77,78]. Thus, two non-synonymous substitutions in the *aristaless-like homeobox 1 transcription factor* (*ALX1*) promoted beak diversification in Darwin’s finches [17]. Interestingly, *aristaless-related homeobox protein* (*arx*) is only expressed in the radula tissue and carries four non-synonymous af-SNPs.

Given similar gene regulatory networks, evolutionarily relevant mutations are expected to accumulate in so-called hotspot genes [82,83]. However, the radula does not share the developmental basis that jaws and beaks have in common [92]. Nonetheless, our observations might be explained by a relatively restricted and highly conserved set of tissue patterning cell–cell signaling pathways [93] that contain a limited set of genes with the potential to rapidly generate adaptive morphological diversity without fatal pleiotropic effects [71,82,83,94]. Although the large number of candidate genes in this study calls for further verification, our results indicate that diversification of foraging organs in adaptive radiations might be achieved via a limited set of cell–cell signaling genes that are particularly prone to rapid adaptive diversification.

## 5. Conclusions

Our study confirms habitat-correlated radula disparity in *T. sarasinorum* and shows evolutionary divergence of ecomorphs. In combination with increased overall sequence divergence and higher proportions of highly significantly DE genes in the radula, these data support the important role of radula adaptation for lineage divergence in adaptive radiations of *Tylomelania*. Finally, overlapping gene sets appear to contribute to rapid adaptive diversification of foraging organs in radiations of fishes, birds, and snails. Since key adaptive traits have primarily been studied in vertebrates, freshwater snails of the genus *Tylomelania* represent an excellent model system to obtain a more general understanding of the genetic mechanisms that generate functional diversity in adaptive radiations.

## Figures and Tables

**Figure 1 genes-13-01029-f001:**
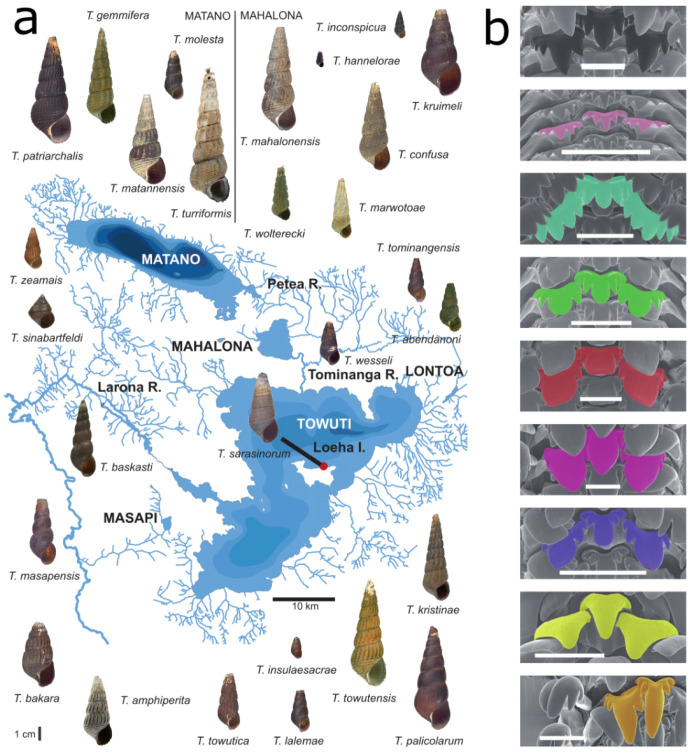
Diversity of *Tylomelania* in the Malili Lakes. Species diversity in the Malili Lakes and surrounding rivers (**a**) is shown together with an overview of radula morphologies (**b**) (Scale bars = 0.1 mm). The sampling site of *T. sarasinorum* at Loeha Island is indicated by a red dot. Modified with permission from [24,25] (8 June 2022, Springer Nature).

**Figure 2 genes-13-01029-f002:**
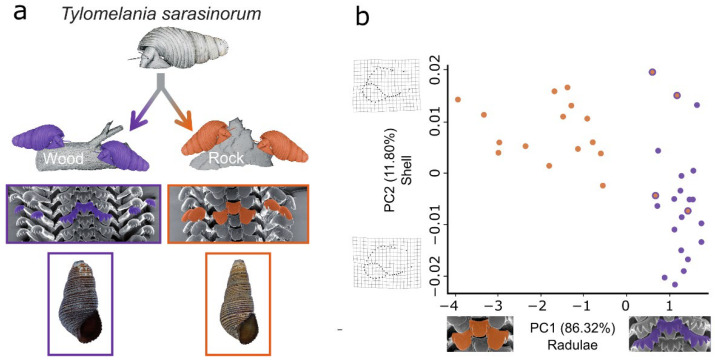
Habitat-correlated radula polymorphism in *T. sarasinorum*. (**a**) Illustrates the hypothesis that radulae of *T. sarasinorum* evolved in adaptation to different microhabitats giving rise to diverging ecomorphs. (**b**) Scatterplot based on the two principal components of shell and radula shape that differed significantly between wood (purple) and rock (orange) ecomorphs. PC1 of radula shape is displayed on the *x*-axis and PC2 of shell shape on the *y*-axis. Thin plate splines visualize the variation in shell shape explained by PC2. The center of each dot indicates the habitat, whereas the outer ring indicates the ecomorph based on SEM inspection of the radula.

**Figure 3 genes-13-01029-f003:**
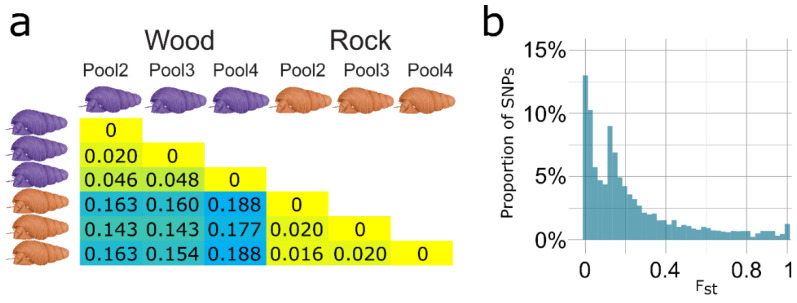
Evolutionary divergence of *T. sarasinorum* ecomorphs. (**a**) Depiction of median SNP-wise F_st_, for pairwise comparisons of pools of both ecomorphs. The degree of differentiation from low to high is indicated by color change from yellow to blue. (**b**) Distribution of SNP-wise F_st_ between ecomorphs for 517,852 putative SNPs (60× minimum coverage (20× per pool), 10% MAF, all pools of each ecomorph combined). Although some SNPs exhibit high differentiation, the majority of variation is shared among both ecomorphs.

**Figure 4 genes-13-01029-f004:**
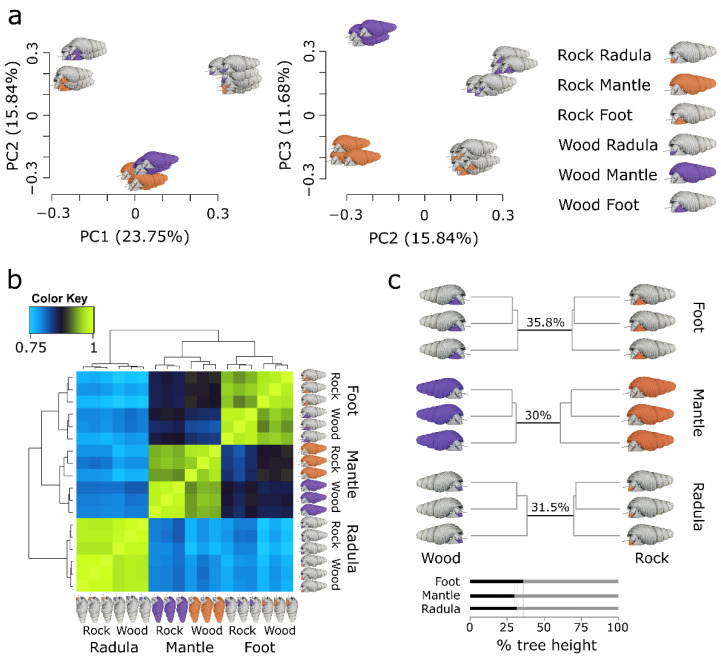
Divergence of gene expression between *T. sarasinorum* ecomorphs. (**a**) Depicts a principal component analysis of gene expression in radula, mantle, and foot tissue from wood (violet) and rock (orange) ecomorphs. The first two principal components (PCs) primarily separate different tissue types, whereas the third PC separates tissues of different ecomorphs. (**b**) Hierarchically clustered Spearman correlation matrix of gene expression (log_2_ transformed counts per million mapped reads). Samples with a more similar gene expression cluster together in the matrix and the hierarchical clustering tree (left and top). Color gradient from blue to yellow shows increasing correlation of gene expression. (**c**) Tree plot illustrating Euclidean distances between samples of each tissue based on expressed genes in that tissue. Relative divergence in overall gene expression between ecomorphs is shown as the branch length separating samples from both ecomorphs (black, bottom).

**Figure 5 genes-13-01029-f005:**
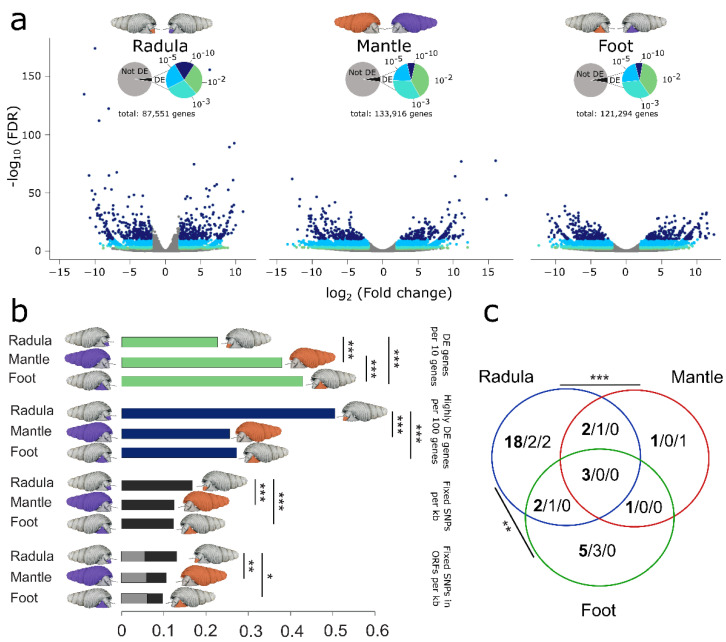
Tissue-wise transcriptomic divergence of *T. sarasinorum* ecomorphs. Volcano plots with log-fold change on x- and the −log_10_ FDR on the *y*-axis illustrate differences in gene expression between ecomorphs (**a**). Pie charts show the proportion of differentially expressed (DE) genes with different FDR. Volcano plots are colored according to pie charts. For genes expressed in each tissue (**b**) shows proportions of DE genes (FDR < 10^−2^), highly DE genes (FDR < 10^−10^), and alternatively fixed SNPs (in ORFs) (black: synonymous; grey: non-synonymous). (**c**) Venn graph illustrating the position of af-SNPs in highly DE genes. The total number is shown first (bold), followed by synonymous and non-synonymous SNPs. In (**b**) and (**c**), significant differences between tissues are indicated by asterisks (* *p* ≤ 0.05; ** *p* ≤ 0.01; *** *p* ≤ 0.001). In (**c**), these refer to differences in highly DE genes with fixed SNPs of any kind.

## Data Availability

Sequence data and additional information are available at the NCBI Sequence Read Archive (SRP134819, SRR10883114–SRR10883125 and SRR6824266–SRR6824274) and BioProject (BioProject ID: PRJNA437798; BioSample accessions: SAMN08685289–SAMN08685300 and SAMN13841508–SAMN13841519).

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
