# Peer review of "Evolutionary Divergence and Radula Diversification in Two Ecomorphs from an Adaptive Radiation of Freshwater Snails"

_genes, 2022, doi:10.3390/genes13061029_

Round 1
Reviewer 1 Report
The work presented by Hilgers et al., in the manuscript titled ‘Evolutionary divergence and radula diversification in two ecomorphs from an adaptive radiation of freshwater snails’ provides a detailed understanding of the role of radula towards ecomorph divergence in T. sarasinorum. The authors exploited morphometry, gene expression metrics and SNP statistics to identify candidate genes that might play a role in radula divergence. The analyses procedures are comparable to the current standards. The methods were detailed enough and the manuscript is well presented. The work has novelty in adding SNP information to identify candidate genes along with the traditional expression counts based procedures. This is an in-depth study supported with well-prepared figures and flow charts.
Line 81: SEM instrument model may be given.
The description of landmarks may be provided in Supplementary Figure 14.
The abbreviations like SEM, PCA, PC, af-SNPs etc. should be expanded at first mention.
Lines 105 – 107: It is stated to have used 19 wood morph specimens and 20 rock morph specimens. The numbers are vice versa as per the supplementary Figure 4. The incorrect one may be corrected.
Line 268-269: How Figure 4a supports this conclusion?
Line 546: Hilgers et al., 2016 is not mentioned in the text.
Generally, PCA is meant for continuous variables. But in this study, PCA was used for the analysis of 2 variables, one is a ratio and the other is a number. The authors should justify the application of PCA for these variables in the study. If any changes were made to the conventional PCA method to suit to the analysis of dis-continuous variables, the same may be given in the manuscript. Otherwise, this limitation may be stated in the manuscript.
Author Response
The work presented by Hilgers et al., in the manuscript titled ‘Evolutionary divergence and radula diversification in two ecomorphs from an adaptive radiation of freshwater snails’ provides a detailed understanding of the role of radula towards ecomorph divergence in T. sarasinorum. The authors exploited morphometry, gene expression metrics and SNP statistics to identify candidate genes that might play a role in radula divergence. The analyses procedures are comparable to the current standards. The methods were detailed enough and the manuscript is well presented. The work has novelty in adding SNP information to identify candidate genes along with the traditional expression counts based procedures. This is an in-depth study supported with well-prepared figures and flow charts.
We would like to thank the reviewer for their interest in our work and for the very helpful and constructive comments.
Comment 1:
Line 81: SEM instrument model may be given.
Response 1:
The instrument model (ZEISS EVO LS 10) is now provided in line 81.
Comment 2:
The description of landmarks may be provided in Supplementary Figure 14.
Response 2:
In addition to visualizing all landmarks, we now provide further descriptions of the landmarks used in the caption of Suppl. Fig 14:
“Suppl. Fig 14: Morphological characterization of T. sarasinorum shell and radula. a) Landmarks (black) and semi-landmarks (red) were positioned on digital photographs of T. sarasinorum shells from individuals collected from wood and rock substrates at Loeha Island. Shell morphology was measured with eight landmarks, i.e., one on the most apical end of the aperture, one on the opposite end of the aperture, at the visual intersection of the aperture with the outside of the first whorl and each one at either side of the shell at the suture of the first and the second whorl, the second and the third whorl, and the fourth and the fifth whorl. Four semilandmarks were used to characterize the aperture, its outer lip and the shape of the first whorl. These consisted of 10 sliding landmarks with the exception of the shortest stretch from the top of the aperture to the top of the first whorl (most bottom right semilandmark). b) Length measurements (blue lines) of the rachis (black box) and its central rachis denticle were used to characterize radula shape of T. sarasinorum wood (purple) and rock (orange) ecomorphs.”
Comment 3:
The abbreviations like SEM, PCA, PC, af-SNPs etc. should be expanded at first mention.
Response 3:
All abbreviations are now expanded at first mention including SEM (line 81), PCA (line 95), PC (line 104), and af-SNPs (line 162).
Comment 4:
Lines 105 – 107: It is stated to have used 19 wood morph specimens and 20 rock morph specimens. The numbers are vice versa as per the supplementary Figure 4. The incorrect one may be corrected.
Response 4:
We thank the reviewer for spotting this error, which is now corrected in Supplementary Figure 4.
Comment 5:
Line 268-269: How Figure 4a supports this conclusion?
Response 5:
We agree with the reviewer that this particular point is primarily captured by Figure 4b and have corrected this figure reference accordingly.
Comment 6:
Line 546: Hilgers et al., 2016 is not mentioned in the text.
Response 6:
In our version of the manuscript Hilgers et al. (2016) appears to be referenced as [28] in line 62.
Comment 7:
Generally, PCA is meant for continuous variables. But in this study, PCA was used for the analysis of 2 variables, one is a ratio and the other is a number. The authors should justify the application of PCA for these variables in the study. If any changes were made to the conventional PCA method to suit to the analysis of dis-continuous variables, the same may be given in the manuscript. Otherwise, this limitation may be stated in the manuscript.
Response 7:
Since we aimed at generating the most comprehensive description of radula tooth shape possible, we included the number of denticles in the principal component analysis. However, since PCA is not recommended for count data, we now reran the analysis excluding the count data. As expected, given the reduced total variation in the dataset, the explained variance of PC1 increased from 86.3% to 93.66%. All reported findings including i) radula morphology clearly distinguishes both ecotypes, ii) differences in radula morphology are more pronounced than differences in shell morphology remain unchanged. To illustrate, we further added the relevant plot from Figure 2b without the count data as Supplementary figure 15.

Reviewer 2 Report
Extensive and detailed research, well written publication - almost nothing to note. Two small remarks:
1. In the results of geometric morphometrics it's shown that radula shape variation (PC1) is 86,3% (the same is on Fig.2), but in the supplement Table 1 the number is 0.42405 (corresponds to 42,4%). I would assume misprint as it's the same number to the next column.
2. I think, it would be useful to separate conclusions from discussion as there are quite a lot of data and hypotheses discussed.
Author Response
Extensive and detailed research, well written publication - almost nothing to note. Two small remarks:
We thank the reviewer for their positive assessment of our work.
Comment 1:
In the results of geometric morphometrics it's shown that radula shape variation (PC1) is 86,3% (the same is on Fig.2), but in the supplement Table 1 the number is 0.42405 (corresponds to 42,4%). I would assume misprint as it's the same number to the next column.
Response 1:
86.3% is correct in the main text. Supplementary table 1 does not refer to radula variation, but shows PCs of shell shape and the proportion of variance explained. The next column shows the cumulative variance explained, which means it displays the sum of that PC and previous PCs. Therefore, the two columns are identical for the first PC.
To avoid misleading references and better explain the use of PCs of both datasets, we modified the potentially misleading positioning of the reference to Suppl. table 1. It now reads: “In comparison, differences in shell shape were less pronounced, as PC2 only explained 11.8% of shell shape variation (Suppl. tables 1, 2) and shell morphospaces of both ecomorphs were overlapping along this axis (Figure 2b)” (lines 225-225).
To further point out the combination of both radula and shell shape datasets in Figure 2b, we added the following explanatory sentence to the caption of Figure 2: “PC1 of radula shape is displayed on the x-axis, and PC2 of shell shape on the y-axis.”
Comment 2:
I think, it would be useful to separate conclusions from discussion as there are quite a lot of data and hypotheses discussed.
Response 2:
We thank the reviewer for this valuable comment and added an additional conclusions paragraph.
“This study confirms habitat-correlated radula disparity in T. sarasinorum and shows evolutionary divergence of ecomorphs. In combination with increased overall sequence divergence and higher proportions of highly significantly DE genes in the radula, these data support an important role of radula adaptation for lineage divergence in adaptive radiations of Tylomelania. Finally, overlapping gene sets appear to contribute to rapid adaptive diversification of foraging organs in radiations of fishes, birds and snails. Since key adaptive traits have primarily been studied in vertebrates, freshwater snails of the genus Tylomelania represent an excellent model system to get a more general understanding of the genetic mechanisms that generate functional diversity in adaptive radiations.”
